# Efficient Pipeline Conflict Resolution for Layered QC-LDPC Decoders in OFDM-PON

Zhijie Wang [1], Zhengjun Xu [1], Kun Chen [1], Yuanzhe Qu [1], Xiaoqun Liu [2], Yingchun Li [1] and Junjie Zhang [1,*]

[1] Key Laboratory of Specialty Fiber Optics and Optical Access Networks, Joint International Research Laboratory of Specialty Fiber Optics and Advanced Communication, Shanghai University, Shanghai 200444, China; zhijie_wang@shu.edu.cn (Z.W.); xuzj@shu.edu.cn (Z.X.); chenkun2001@shu.edu.cn (K.C.); quyuanzhe@shu.edu.cn (Y.Q.); liyingchun@shu.edu.cn (Y.L.)

[2] Beijing Smartchip Microelectronics Technology Co., Ltd., Beijing 100000, China; liuxiaoqun_mail@foxmail.com

* Correspondence: zjj@staff.shu.edu.cn

**Abstract:** The high standard of communication quality in optical access networks makes forward error correction (FEC) schemes, such as LDPC, an integral part of the system. However, pipeline conflict arising from data dependencies is a common issue encountered in the hardware implementation of layered QC-LDPC decoders. This paper proposes an efficient layered decoding architecture to reduce pipeline conflicts without introducing stall cycles. It can solve some of the pipeline conflicts by flexibly reordering the processing order of inter-layer and intra-layer submatrices offline. In addition, the patch method, based on variable-to-check messages, allows for the delayed use of gains between layer iterations, which can further minimize the performance loss caused by the remaining pipeline conflicts. The experimental results on the LDPC code of the IEEE802.16 standard in the OFDM-PON system demonstrate that the proposed architecture has sensitivity improvements of 0.125 dBm and 0.375 dBm, respectively, compared with our previous work and the method described in the other work. The optimized architecture improves the reliability of the decoder and can also make a contribution to efficient PON systems.

**Keywords:** QC-LDPC; pipeline conflicts; patch method; variable-to-check message; OFDM-PON





## 1. Introduction

The explosive growth of bandwidth-intensive services such as ultra-high-definition live streaming, Cloud Office applications, and 5G technology has posed further challenges to Quality of Service (QoS) [1] and Dynamic Bandwidth Allocation (DBA) [2], and other aspects of optical access networks. Various Passive Optical Network (PON) technologies have emerged to address these challenges, and the characteristics and applicable scenarios of each technology are shown in Table 1. Orthogonal Frequency Division Multiplexing Passive Optical Access Network (OFDM-PON) [3] has garnered significant attention in metropolitan and backbone networks owing to its high spectral efficiency and robust interference resistance, particularly in scenarios involving ultra-large capacity and long-distance transmission. However, during high-speed data transmission, signal attenuation and noise interference are unavoidable issues. To further bolster transmission reliability, the integration of forward error correction (FEC) codes can be considered.

The low-density parity-check (LDPC) code exhibits the remarkable property of approaching the Shannon limit, making it one of the most promising forward error correction codes currently available [4]. Thus, it is widely used in various fields, including deep-space communications [5], storage devices [6], and 5G NR [7]. Many works have also introduced its practical applications in optical access networks [8–11]. LDPC codes can be classified into random code, cyclic code, and quasi-cyclic code (QC-LDPC) based on their type characteristics. The original LDPC code is random, and its parity-check matrix (PCM) rarely has structural characteristics, leading to highly complex encoding and decoding procedures. In

contrast, the cyclic code has obvious structural characteristics, which can greatly simplify the hardware implementation process. However, the existing cyclic code typically has a large row weight, making the decoder implementation complex and subtle. QC-LDPC is composed of zero submatrices of the same size and cyclic shift submatrices, which offer favorable structural characteristics and can simplify the design of encoders and decoders. Therefore, QC-LDPC is currently a popular choice for the OFDM-PON system.

**Table 1.** Pros and cons of various PONs.

| PON Technology | Pros/Cons | | Application Scenarios | |
| --- | --- | --- | --- | --- |
| TDM-PON | - | Cost-effective and mature technology. | - | Residential broadband access. |
| | - | Easy to deploy and maintain. | | |
| | - | Limited bandwidth per user. | | |
| | - | Limited scalability. | | |
| WDM-PON | - | High bandwidth and scalability. | - | Large enterprises. |
| | - | Supports multiple users with dedicated bandwidth. | - | Data centers. |
| | - | More complex and expensive. | | |
| | - | Requires sophisticated network management. | | |
| OFDM-PON | - | High spectral efficiency and resistance to interference. | - | Metropolitan area network. |
| | - | Suitable for long-distance and high-capacity transmission. | - | Backbone network. |
| | - | Complex implementation and hardware requirements. | | |
| | - | Potential frame synchronization issues. | | |
| | - | High Peak-to-Average Power Ratio. (PAPR). | | |

Among numerous decoding algorithms, the layered decoding schedule is more popular due to its superior decoding convergence [12]. During the hardware implementation process, the quasi-cyclic characteristic of the QC-LDPC code is highly suitable for partially parallel-layered decoding architectures [13]. This approach can achieve a balance between hardware resource consumption and throughput. However, certain QC-LDPC decoders aim to further improve throughput by increasing the system's operation frequency, which can be achieved by inserting more pipeline stages. However, the presence of data dependencies in the iterative decoding process across contiguous layers poses a challenge, as the introduction of pipelines significantly exacerbates the issue of data updating conflicts in message memory [14]. When the next layer is ready to utilize the updated messages, the corresponding update process for the previous layer will not yet have been completed. This dilemma is recognized as the pipeline conflict problem.

*1.1. Related Works*

Traditional processing schemes merely introduce multiple stall cycles during the processing stage and wait for the completion of the update of log-likelihood ratios (LLRs) at the previous layer before proceeding to the next layer. The number of stall cycles inserted is often substantial, leading to a significant reduction in decoding throughput.

In [15,16], partial pipeline conflicts can be reduced by rearranging the processing order of the submatrix using techniques such as graph coloring or solving the traveling salesman problem. Similarly, Ref. [17] proposes the method of combining block scheduling and inserting no-operation instructions to resist the occurrence of conflicts. Although these methods can reduce the corresponding stall cycles, complete elimination is not achievable. To completely eliminate the stall cycles, the method from [18] employs a flooding decoding mechanism when pipeline conflicts occur, while using a layered mechanism at other times.

However, this approach incurs a significant performance loss compared to the theoretical layered decoding algorithm, necessitating more iterations. Similarly, the residue-based layered schedule [19] continuously accumulates and stores the contributions in registers when pipeline conflicts arise, adding them to the corresponding LLRs at the end of the pipeline problem. Ref. [20] proposes to split the matrix for the LDPC codes in the DVB-T2 protocol, which can reduce the degree of parallelism and significantly reduce the number of conflicts. The remaining conflicts are avoided by adding the equivalent matrix of puncture sites. However, decreased parallelism translates to lower throughput. Ref. [21] proposed a dynamic planning method for node reordering within layers, but the read and write order consistency problem brings about a limitation of reordering.

In our previous work [22], we introduced a priority-based layered decoder with a double update queue. The occurrence of conflicts is reduced by reordering the submatrix processing within each layer. Additionally, processing is divided into overlapping and non-overlapping paths in parallel according to the arrangement order, greatly reducing the processing delay. For unavoidable conflicts, corresponding gains will be calculated and added to recent usage to minimize performance losses. However, when the number of pipeline stages inserted into the decoder exceeds the maximum check node weight, the double update queue will be completely invalidated.

### 1.2. Overview and Contribution

To address pipeline conflicts efficiently, we introduce an optimized QC-LDPC layered decoder without inserting stall cycles. The main contributions are:

1. A decoding method based on a patched variable-to-check message is proposed. When necessary, it is preferable to read the un-updated LLR and apply a patch to the variable-to-check message, rather than waiting to read the updated LLR. This approach effectively reduces pipeline conflicts.
2. A more flexible rearrangement of the inter-layer and intra-layer submatrix processing order is allowed in the proposed hardware architecture. It effectively eliminates pipeline conflicts caused by overlapping submatrices among three or more successive layers of traditional decoding.
3. The proposed decoding architecture is implemented on hardware and the performance improvement is demonstrated experimentally on the OFDM-PON platform. The experimental results demonstrate that the proposed architecture has a performance improvement of 0.125 dBm compared to our previous work [22] and 0.375 dBm over the residual-based decoder in the literature [19] under the maximum 10 iterations of decoding and a 64-QAM modulation format.

The remaining part of this paper is structured as follows: Section 2 briefly reviews the layered decoding algorithm and the pipeline conflict encountered in hardware implementation. Section 3 delves into the detailed design of the proposed decoder. Section 4 presents the experimental setup on the OFDM-PON platform and analyses the results. Finally, Section 5 provides a concise summary of the paper.

## 2. Conflict Problems in Pipelined Layered Decoders

### 2.1. Layered Decoding Algorithm

Layered decoding is adopted by most practical systems because each layer can utilize the latest a posteriori (AP) LLR when updating, which shows better decoding convergence performance.

For the layered decoding algorithm, each row of the check matrix $H_{(N-K) \times N}$ can be viewed as a component code, and each component code shares the same variable node information. Each component code is decoded in turn, and each decoding process includes three steps: variable node update, check node update, and LLR update.

In an additive white Gaussian noise (AWGN) channel, the initialization of LLR of the variable node $V_i$ is given as

$$LLR_v^{init} = log \frac{P(x_v = 0|y_v)}{P(x_v = 1|y_v)} = \frac{2y_v}{\sigma^2} \tag{1}$$

where $x_v$ represents the transmitted bit of variable node $V_i$, $\sigma^2$ represents the noise variance, $P(x_v = x|y_v)$ represents the probability that $x_v$ is equal to the value $x$ ($x = 0$ or $x = 1$), and $y_v$ represents the received symbol.

The check-to-variable message ($L_{i \rightarrow j}$) is initialized to 0.

$$L_{i \rightarrow j}^{(0,i)} = 0 \tag{2}$$

During the update process of the *i*-th layer in the *it*-th iteration, the variable-to-check messages ($L_{j \rightarrow i}$) are calculated using the check-to-variable message ($L_{i \rightarrow j}$) from the previous ($it - 1$)-th iteration and the a posteriori probability LLR.

$$L_{j \rightarrow i}^{(it,i)} = LLR(j) - L_{i \rightarrow j}^{(it-1,i)} \tag{3}$$

When all the variable-to-check messages ($L_{j \rightarrow i}$) in a layer are available, a check-to-variable message ($L_{i \rightarrow j}$) can be generated with MSA by the following Equation (4):

$$L_{i \rightarrow j}^{(it,i)} = \prod_{j' \in V_i \setminus j} sgn\left(L_{j' \rightarrow i}^{(it,i)}\right) \cdot max\left(\min_{j' \in V_i \setminus j}\left(|L_{j' \rightarrow i}^{(it,i)}|\right)\right) \tag{4}$$

Although MSA reduces the complexity of hardware implementation, it causes a degradation in decoding performance. To compensate for the loss of performance, the magnitude of the check-to-variable message ($L_{i \rightarrow j}$) can be multiplied with a normalization factor $\alpha$ or have an offset factor $\beta$ subtracted, processes which are referred to as the normalized min-sum algorithm (NMSA) and offset min-sum algorithm (OMSA), respectively, as proposed in (5) and (6).

$$L_{i \rightarrow j}^{(it,i)} = \alpha \cdot \prod_{j' \in V_i \setminus j} sgn\left(L_{j' \rightarrow i}^{(it,i)}\right) \cdot \min_{j' \in V_i \setminus j}\left(|L_{j' \rightarrow i}^{(it,i)}|\right) \tag{5}$$

$$L_{i \rightarrow j}^{(it,i)} = \prod_{j' \in V_i \setminus j} sgn\left(L_{j' \rightarrow i}^{(it,i)}\right) \cdot max\left(\min_{j' \in V_i \setminus j}\left(|L_{j' \rightarrow i}^{(it,i)}|\right) - \beta, 0\right) \tag{6}$$

The LLR for variable node $V_i$ can be updated once the corresponding check-to-variable message ($L_{i \rightarrow j}$) has been calculated, as given by (7):

$$LLR(j) = L_{j \rightarrow i}^{(it,i)} + L_{i \rightarrow j}^{(it,i)} \tag{7}$$

When an iteration is complete, the codeword is judged with the soft decision based on the sign of LLR. The decided codeword *C* and the parity-check matrix *H* are used to calculate the syndrome $S = C \times H^T$. If $C \times H^T = 0$ or the decoding has reached the set maximum number of iterations, then this decoding will end.

To sum up, the decoding pseudo code based on the Layered OMSA algorithm used in this article is shown in Algorithm 1. In the algorithm, the length of the LDPC code is *N* and the length of the information bit is *K*. The initialized LLR obtained from the channel is referred to as the *Channel LLR*. The maximum number of iterations is defined as $it_{max}$.

---

**Algorithm 1.** Algorithm of Layered OMSA

---

**Initialization:**
    set $LLR(j)(j = 1, 2, \ldots, N)$ to *Channel LLR*
    set $L_{i \to j}^{(0,i)}$ to 0
    set *it* and *S* to 1
**While** $(it > it_{max})$ or $(S = 0)$
    **For** it = 1 : $(N - K)$
        **For** j $\in V_i$
            $L_{j \to i}^{(it,i)} = LLR(j) - L_{i \to j}^{(it-1,i)}$
        **End for**
        **For** j $\in V_i$
            $L_{i \to j}^{(it,i)} = \prod_{j' \in V_i \setminus j} sgn\left(L_{j' \to i}^{(it,i)}\right) \cdot \max\left(\min_{j' \in V_i \setminus j}(|L_{j' \to i}^{(it,i)}|) - \beta, 0\right)$
            $LLR(j) = L_{j \to i}^{(it,i)} + L_{i \to j}^{(it,i)}$
        **End for**
        Hard decision: $C = -sign(LLR)$
        Compute: $S = C \times H^T$
        $it = it + 1$
    **End for**
**End while**

---

### 2.2. Pipeline Conflict Problem

Figure 1 shows an example of a QC-LDPC PCM consisting of $4 \times 9$ cyclic submatrices. The grid boxes represent the zero matrices, while the other boxes represent the matrices offset according to the standard matrix. To fully leverage the cyclic characteristics of submatrices in QC-LDPC, partially parallel layered decoders with parallelism equal to the submatrix size are commonly employed. In hardware implementation, incorporating pipeline operations is an effective approach to enhance the system's operating frequency.

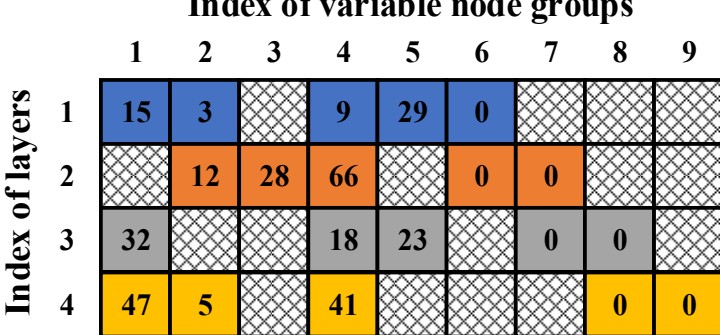

**Figure 1.** An example of a base graph matrix schematic diagram.

According to Section 2.1, the layered decoder algorithm can be divided into several main steps based on the processing sequence. These steps include obtaining LLR, updating variable-to-check messages, updating check-to-variable messages, and updating LLR. Usually, several pipeline stages are inserted between these steps. As shown in Figure 2, reading LLRs from the hardware storage space and writing updated LLRs back into the corresponding storage space are the first and last stage of the pipeline, respectively. In theory, the previous layer should provide updated values for the next layer. The corresponding submatrix of the lines in the diagram shows that the next layer needs to read the update value, but the update value in the previous layer has not been calculated. This situation leads to the pipeline conflict that we mentioned above.

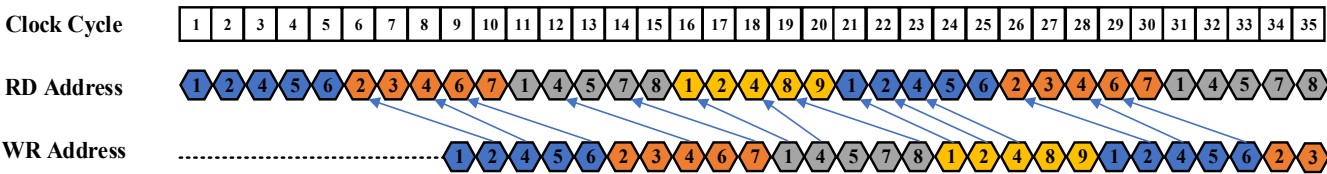

**Figure 2.** The conflicts of inter-layer memory access.

## 3. A Reordered QC-LDPC Decoder with Patched Variable-to-Check Message

### 3.1. Inter-Layer and Intra-Layer Processing Scheduling

In essence, the pipeline conflict problem is caused by the overlap of non-zero submatrices in the PCM. Therefore, the primary approach to resolving pipeline conflicts is to minimize the number of overlapping submatrices. According to the principle of layered decoding, the adjustment of the processing order between layers does not affect the decoder's performance. The number of overlapping submatrices can be reduced by adjusting the processing order between layers properly. Especially for those PCMs with sparse non-zero submatrices, reasonable adjustments of the processing order between the layers can even completely eliminate pipeline conflicts. Figure 3 displays the check matrix of the WiMAX QC-LDPC code with a rate of 1/2. By adjusting the processing order between layers to 0, 2, 4, 11, 6, 8, 10, 1, 3, 5, 7, 9, the elimination of overlapping submatrices in the check matrix can be observed.

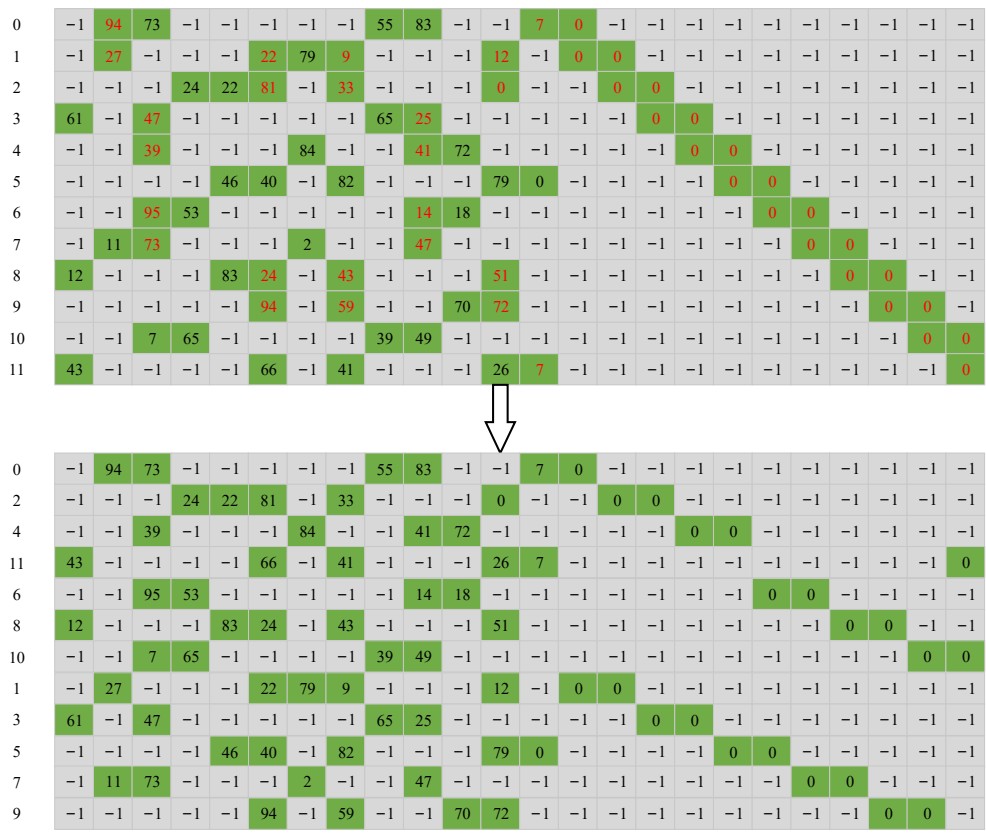

**Figure 3.** The inter-layer reordering for the LDPC check matrix of the WiMAX protocol with a 1/2 rate.

The intra-layer processing order can also be rearranged to reduce pipeline conflicts, in addition to the inter-layer processing order. Two main processes are involved in a layer's processing sequence: reading the LLRs from the memory and writing the updated LLRs back to the memory.

During the reading process, it is preferential to start with the non-overlapping submatrix between the current layer and the preceding layer, followed by the overlapping submatrix.

Conversely, in the write-back process, priority is given to writing back the overlapping submatrix between the current layer and the subsequent layer, followed by the non-overlapping submatrix.

This allows the LLR of the overlapping submatrix to be updated at an earlier time, and to be read and used at a later time. It can minimize the occurrence of pipeline conflicts.

Similar to other works, such as the hardware implementation of layered decoders in [18,19,22], due to the fact that many intermediate process variables are stored in First In First Out (FIFO), the read order and write-back order of each layer of LLR random access memory (RAM) must be consistent. This causes some contradictions in the rearrangement of the reading and writing order of multi-layer overlapping submatrices. In Figure 1, there is an overlapping submatrix (specifically, the fourth submatrix) that exists at the same position across consecutive first, second, and third layers. When analyzing the sequential arrangement of LLR reads and writes for this submatrix in the second layer, several considerations arise. Firstly, since the first layer supplies updated LLRs for the second layer, it is desirable to arrange the reading order of the submatrix's LLR in the second layer to occur later, allowing sufficient time for the updated LLR to be read. Secondly, as the second layer provides updated values for the third layer, there is a need for the LLR of the submatrix in the second layer to be updated promptly, necessitating an earlier write-back order. Consequently, the reading and writing order of the fourth submatrix in the second layer presents a contradiction: the need for a delayed read to accommodate updated LLR from the first layer versus the need for an early write-back to ensure timely updates for the third layer. This conflict highlights the complexities involved in optimizing the reading and writing orders of overlapping submatrices in layered decoders.

In our proposed hardware architecture, it is reinforced that each layer has a different order of reading and updating LLRs. This approach effectively reduces the pipeline conflicts caused by the above situation. Details of the specific hardware structure are provided in Section 3.3.

### 3.2. Patch Method Based on Variable-to-Check Message

Some pipeline conflicts can be avoided by rescheduling the processing order of inter-layer and intra-layer submatrices. However, the remaining ones will still cause significant performance loss. Especially for those PCMs with dense non-zero submatrices, pipeline conflicts are severe, and rescheduling is basically ineffective.

By combining Formulas (7) and (3), the updated formula for LLR can be rewritten as follows:

$$
\begin{aligned}
\mathrm{LLR}^i(\mathrm{j}) \quad &= L_{j \to i}^{(it,i)} + L_{i \to j}^{(it,i)} \\
&= (\mathrm{LLR}^{i-1}(\mathrm{j}) - L_{i \to j}^{(it-1,i)}) + L_{i \to j}^{(it,i)} \\
&= \mathrm{LLR}^{i-1}(\mathrm{j}) + \left( L_{i \to j}^{(it,i)} - L_{i \to j}^{(it-1,i)} \right) \\
&= \mathrm{LLR}^{i-1}(\mathrm{j}) + \Delta L_{i \to j}^i
\end{aligned}
\tag{8}
$$

where $\Delta L_{i \to j}^i$ is defined as residue or gain, representing the difference before and after the update of the check-to-variable messages. It reflects the decision information newly contributed by the check node during the iteration process. Therefore, in the layered decoding algorithm, it effectively obtains new gains from the check node through continuous layer traversal.

To achieve the ideal pipeline effect, when pipeline conflicts occur, the only option is to read the un-updated LLRs. However, this results in the loss of the gain generated by the corresponding check node, which cannot positively impact subsequent iterations. Referring to Formula (8), many references [18,19,22] suggest that, in such cases, the corresponding gain can be calculated and stored in the relevant memory space. The gain will not be read and stacked until the LLR has been updated at the same position. This ensures that the gain

is only temporarily disregarded during conflicts but will be incorporated in the subsequent iteration process, thereby positively contributing to convergence acceleration. This type of decoding architecture is referred to as the patch method based on LLR.

However, shifting the patching position from the LLR to the variable-to-check message reveals that reading the un-updated LLR may not necessarily induce pipeline conflicts. This adjustment can further reduce pipeline conflicts. We refer to this proposed architecture as the patch method based on variable-to-check message. As Formula (4) is rewritten, the variable-to-check messages will be updated to

$$L_{j \to i}^{(it,i)} = \left( LLR^{old}(j) - L_{i \to j}^{(it-1,i)} \right) + \Delta L_{i \to j}^{i-1} \tag{9}$$

From a different perspective, for those overlapping submatrices where the gain of the previous layer has already been calculated before calculating the variable-to-check messages in the next layer, there will be no pipeline conflicts caused by reading the old LLRs, because we replace the reading of updated LLRs with reading old LLRs and patching variable-to-check messages. As illustrated in Figure 4, in our proposed architecture, although the un-updated LLRs were read, no pipeline conflicts were generated.

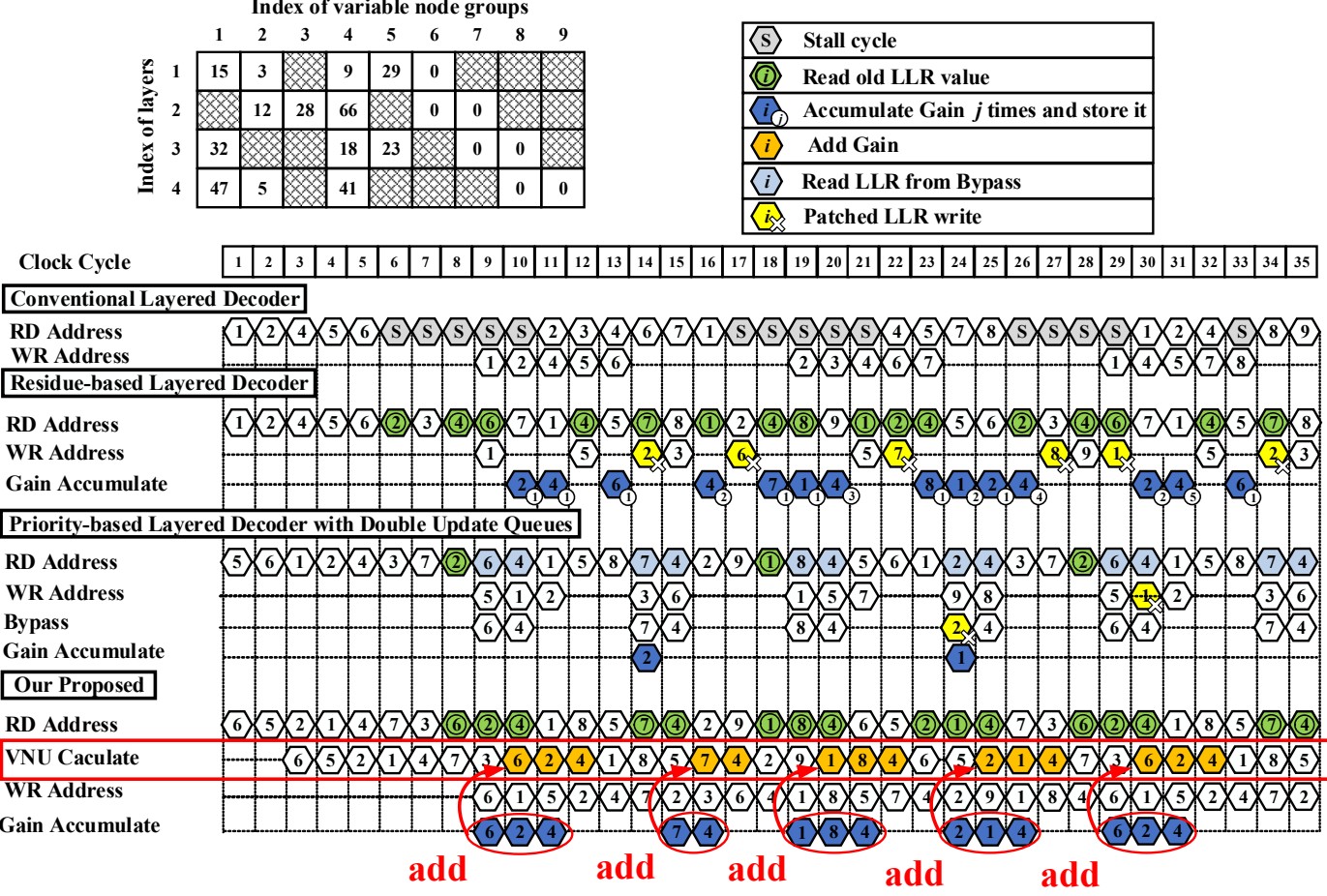

**Figure 4.** Comparison diagram between different decoding architectures.

Figure 5 shows the main process flow charts of the patch method based on LLR and the patch method based on the variable-to-check message. The letters on the arrow lines represent the number of pipeline stages inserted in the hardware implementation.

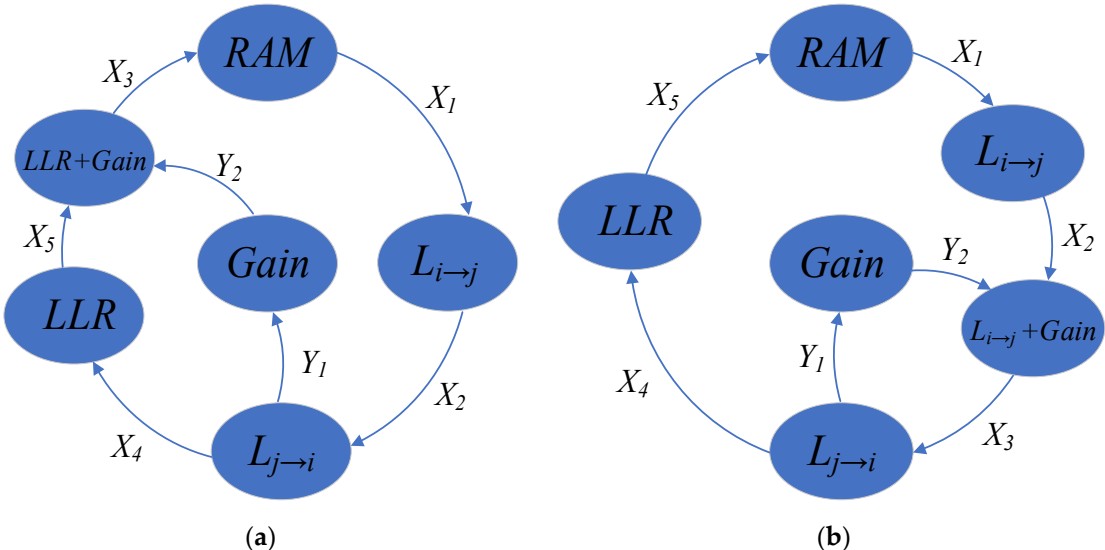

**Figure 5.** The main process flow charts of the patch method based on LLR and variable-to-check message. (**a**) The patch method based on LLR; (**b**) The patch method based on variable-to-check message.

It is assumed that there is an overlapping non-zero submatrix between two adjacent layers, and the moments when the LLR is read by the two layers are $N_1$ and $N_2$, respectively.

For the patch method based on LLR, when there is no pipeline conflict, $N_1$ and $N_2$ need to meet the following requirements:

$$N_2 \geq N_1 + X_1 + X_2 + X_3 + X_4 + X_5 \tag{10}$$

For our proposed scheme, we only need to ensure that the gain of the first layer is calculated before the second layer finishes calculating the variable-to-check message. Therefore, we need to meet the following:

$$N_2 + X_1 + X_2 \geq N_1 + X_1 + X_2 + X_3 + Y_1 + Y_2 \to N_2 \geq N_1 + X_3 + Y_1 + Y_2 \tag{11}$$

Since $(Y_1 + Y_2)$ is always less than $(X_1 + X_2 + X_4 + X_5)$ during the design process, the condition for the patch method based on LLR to avoid pipeline conflicts is more stringent than that for the patch method based on variable-to-check messages. When the number of pipeline stages and overlapping submatrices is sufficiently high, the patch method based on variable-to-check messages can reduce the number of pipeline conflicts by $(X_1 + X_2 + X_4 + X_5 - (Y_1 + Y_2))$ in each layer iteration compared to the patch method based on LLR.

### 3.3. Proposed Hardware Implementation Structure

Figure 6 shows the hardware design details of the layered decoder that we propose. The decoder mainly consists of the LLR RAM, Variable Node Unit (VNU), Check Node Unit (CNU), LLR update Unit (LU), Gain generate Unit (GU), other auxiliary units (barrel shifter, cache, etc.), and control modules. The parallelism of the decoder is equal to size Z of the submatrix, which means that Z VNUs, CNUs, LUs and GUs will work simultaneously.

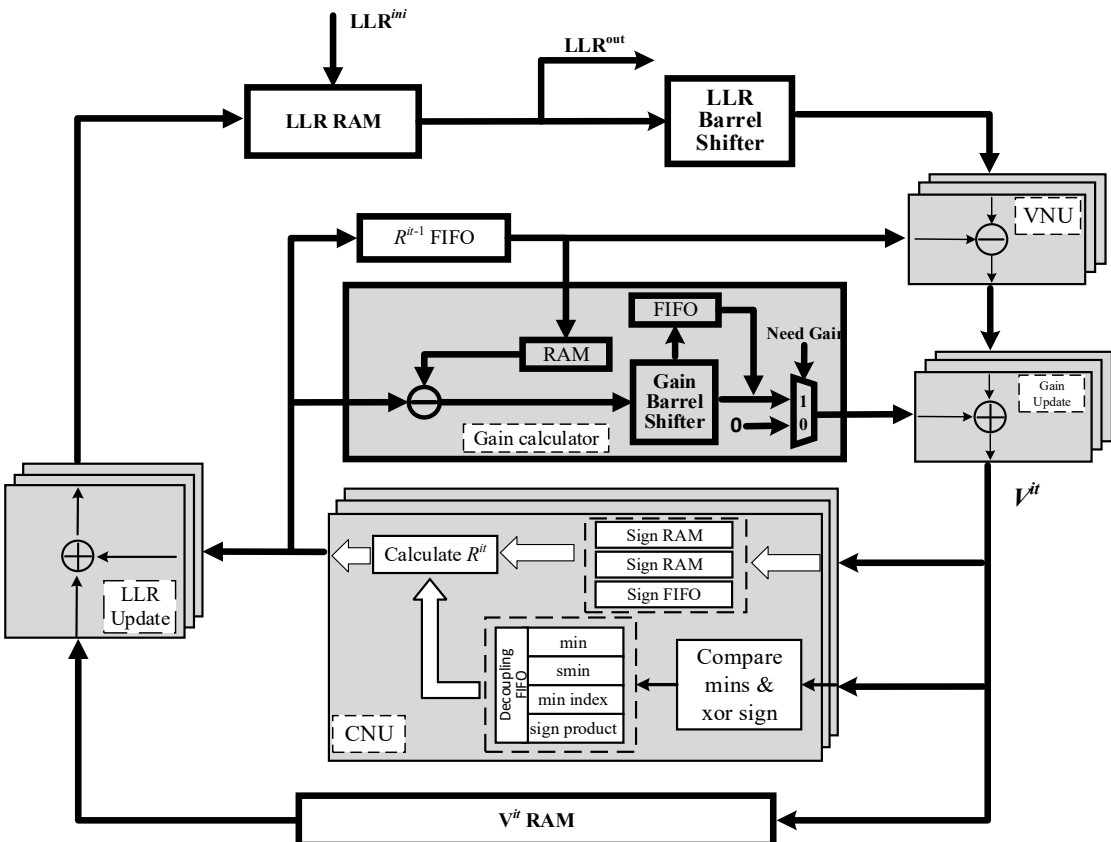

**Figure 6.** Proposed hardware architecture of the layered QC-LDPC decoder.

Initially, the initialized channel LLRs are written into LLR RAM. The depth of LLR RAM is the number of recurrent submatrices in a layer, and each storage space contains LLR information for Z variable nodes.

When decoding begins, the LLRs of the corresponding addresses in RAM are sequentially read and fed into the barrel shifter according to the rearrangement order of the inter-layer and intra-layer submatrix processing in Section 3.1. The barrel shifter is responsible for adjusting the order of information during decoding.

In the VNU, the variable-to-check messages are obtained by subtracting the check-to-variable messages of the previous iteration from the LLRs. If the LLRs previously read are not up-to-date and the corresponding gain of the last layer can be obtained at this moment, then the variable-to-check messages obtained here need to be patched with gain. At this point, it is still believed that no pipeline conflict has occurred. Only when the gain of the last layer lags behind the end time of variable-to-check message calculation will it be considered that pipeline conflict has truly occurred. And the gain will be cached until the next time, when the same variable-to-check message is recalculated, and patches will be applied to compensate for the loss caused by reading the un-updated LLR. The updated variable-to-check messages are fed into the CNU. On the other hand, they are cached and read out for computation during LLR updates. But unlike other architectures, the hardware used for caching here is RAM rather than FIFO, as there is a need for different write and read orders.

In the CNU, variable-to-check messages are separated into sign bits and amplitudes. By comparing the amplitudes, the minimum and second minimum values in a layer can be obtained, along the positional index corresponding to the minimum value. Additionally, the sign product of all variable-to-check messages in the same layer is calculated. Afterwards, by comparing the positional index of the submatrices that need to be processed with the position index of the minimum value, one can determine whether to use the minimum value or the second minimum value as the amplitude of the updated check-to-variable

messages (because the OMSA algorithm is used, further amplitude processing is required for the minimum and second minimum values). The signs for check-to-variable messages are determined by the sign product and the signs of the corresponding variable-to-check messages. There are three directions for the updated check-to-variable messages: one is directly used to add the updated LLR to variable-to-check messages. Another is cached into the FIFO for the next iteration of the variable-to-check messages update. The rest is used to generate gains to compensate for the loss caused by reading un-updated LLRs. The check-to-variable messages in these three directions need to be calculated separately due to the difference in their order. Therefore, the signs of the variable-to-check messages are originally cached in three separate memory spaces.

In the GU, the gains of the overlapping submatrices are obtained by subtracting the check-to-variable message of the current iteration and the previous iteration. These gains are aligned by barrel shifters and then patched into the corresponding variable-to-check messages.

In the LU, the updated LLRs are obtained from variable-to-check messages and check-to-variable messages. Then, they are rewritten into the LLR RAM.

## 4. Results and Analysis

### 4.1. Experimental Setup

To verify the performance of the layered LDPC decoder with the proposed architecture in OFDM-PON, experiments are undertaken by an FPGA-based OFDM-PON platform previously used in [23–27] and an FPGA-based LDPC decoder, as shown in Figure 7. In this paper, two Xilinx FPGA boards (ML605) with Virtex-6 XC6VLX240T are used in the implementation of the OFDM-PON platform. Table 2 summarizes the key parameters of the platform. To maximize the throughput and operating frequency, another Xilinx FPGA board (VC709) with xc7vx690t is independently used for the LDPC decoder.

For the off-line OFDM transmitter, the binary sequence generated by the pseudo-random binary sequence (PRBS) is multiplied with the LDPC generation matrix G to obtain the encoded data. In the experiment, we used the LDPC code with rate 3/4 of IEEE 802.16 standard [28], and the codeword had a length of 2304 bits. The scatter diagram of the parity-check matrix H is shown in Figure 8.

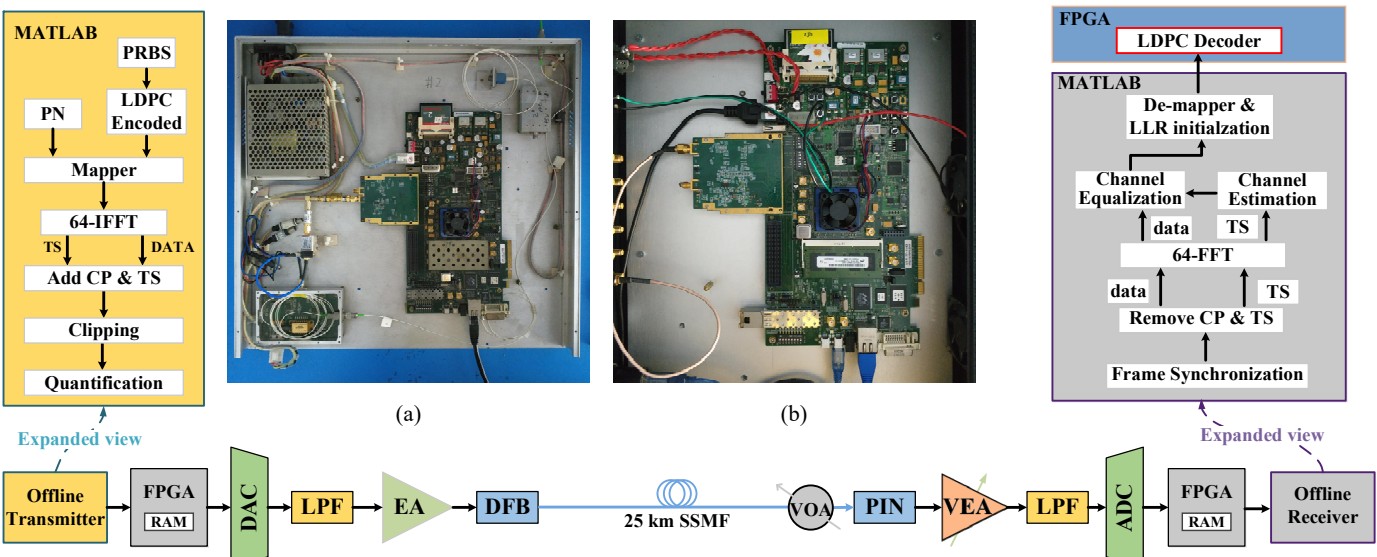

(a)                                                                        (b)

**Figure 7.** Experimental system and transceiver DSP block of the off-line OFDM-PON and FPGA-based LDPC decoder. (**a**) The hardware of transmitter; (**b**) the hardware of receiver.

**Table 2.** OFDM-PON system and LDPC decoder parameters.

| Parameter | Value |
|---|---|
| FFT/IFFT points | 64 |
| Data-carrying subcarriers | From 2 to 28 |
| Modulation format | 16-QAM/64-QAM |
| ADC/DAC resolution | 10/12-bit |
| ADC and DAC sample rate | 4 GS/s |
| OFDM frame CP | 16 samples (4 ns) |
| Transmitter output power | +7.75 dBm |
| DFB wavelength | 1549.98 nm |
| DFB modulation bandwidth | 2.7 GHz |
| DFB bias current | 45 mA |
| DFB driving voltage | 2 Vpp |
| PIN detector bandwidth | 40 MHz~3 GHz |
| PIN responsivity | 0.9 mA/mW |
| Standard | 802.16 |
| Code rate | 3/4 |
| Code length | 2304 |
| Size of submatrices | 96 × 96 |
| Parallelism | 96 |

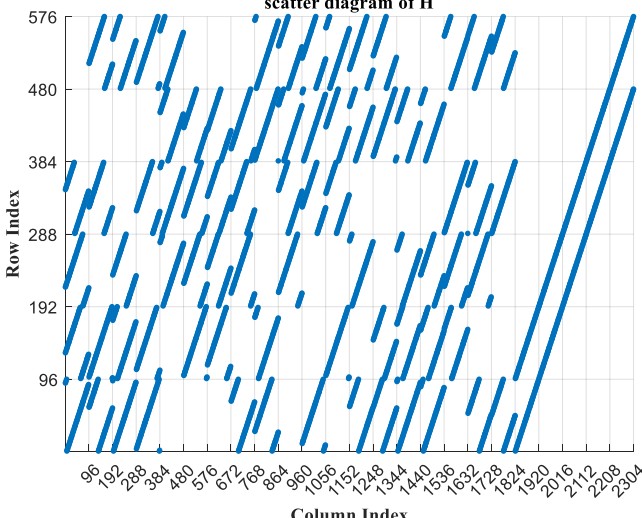

**Figure 8.** The scatter diagram of the LDPC check matrix H with rate of 3/4 in IEEE 802.16 standard.

The encoded data are first converted from a serial signal to a parallel signal and subsequently mapped onto OFDM subcarriers according to the modulation format. There are 64 subcarriers in our OFDM system. However, in order to obtain real-valued IFFT outputs later, all subcarriers need to meet Hermitian symmetry. Therefore, only half of the subcarriers can be used for transmission. Further, to overcome the low-pass characteristics of the channel and the ADC roll-down effects, we only use 2 to 28 subcarriers to carry the encoded data, as we did in our previous work [23]. And the power spectral density when 2 to 28 subcarriers are turned on is shown in Figure 9. Thus, the encoded data are actually converted into 27 parallel data channels.

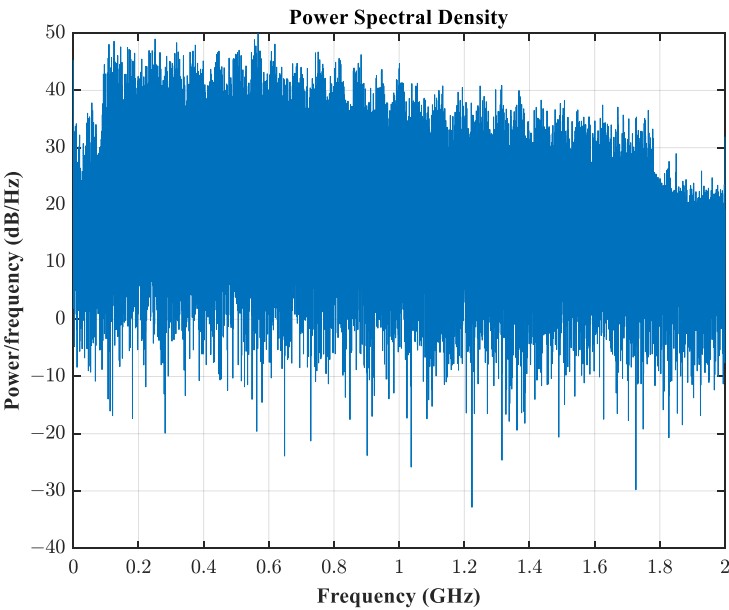

**Figure 9.** The power spectral density when 2 to 28 subcarriers are turned on.

Afterwards, the data are subjected to a 64-point IFFT operation to form M OFDM symbols. The optimization of the IFFT/FFT operation can be referred to in our previous work [27]. Since we designed a complete LDPC codeword to be carried by a single OFDM frame, the size of M was determined by the code length (2304) and the number of valid bits ($27 \times n$, n is 4 for 16-QAM or 6 for 64-QAM) carried by a single OFDM symbol. Then, a 16-sample cyclic prefix (CP) was inserted for each OFDM symbol, and a synchronization header consisting of 80 zeros and 2 ones as well as a training sequence (TS) of 2 FFT sizes were inserted for each OFDM frame. A complete OFDM frame is shown in Figure 10.

| Synchronization Header | | TS_CP | | TS | TS | OFDM Symbol M×80 | | | |
|---|---|---|---|---|---|---|---|---|---|
| Leading zeros (80) | Sync-header (2) | 16 | 16 | 64 | 64 | 80 | 80 | ...... | 80 |

| CP | DATA |
|---|---|
| 16 | 64 |

**Figure 10.** The OFDM frame structure.

The generated OFDM frames are sent to the Xilinx ML605 FPGA board via UDP protocol and stored in the internal BRAMs of the FPGA board. The analog signal is outputted by a 4GS/s DAC with a resolution of 12 bits and directed through a 2 Vpp variable attenuator and a 13 dB amplifier to a narrow bandwidth distributed feedback laser (DFB-LD). The resulting optical signals are then injected into a fiber optic link consisting of 25 km of standard single-mode fiber (SSMF) for transmission.

At the receiver, a variable optical attenuator is used to adjust the received optical power. The optical signal is then converted into an electrical signal by a 2.7 GHz photodiode (PIN) detector. The resulting electrical signal is first amplified by a variable amplifier to ensure that the signal occupies the full dynamic range of the 10-bit ADC. The ADC samples and quantizes the analog signal, converting it into digital signal. Then, the digital signal is de-multiplexed into 32 parallelisms and buffered into a BRAM.

The sampled digital signal then enters the DSP of the receiver. The frame synchronization is first performed by the synchronization header, and the specific synchronization steps can be referred to in our previous work [24]. After the synchronization has been completed, the CP of 16 samples can be removed from all OFDM symbols, followed by a

64-point FFT being performed to obtain the TS and OFDM data. After channel parameters are obtained using TS for channel estimation, the data are subjected to channel equalization and the initialized LLRs are computed based on the modulation format.

These LLRs quantized to 8-bit are sent to the VC709 FPGA. The decoding is caried out according to the proposed layered decoder and other architectures to be compared. After the decoding has been carried out, the corresponding BER and the average number of iterations are counted for analysis and discussion.

### 4.2. Schedule Optimization Results

To demonstrate our proposed schedule optimization results clearly, we broke down the proposed method into four points that are beneficial for reducing pipeline conflicts:

1. Reordering the processing of the inter-layer.
2. Reordering the processing of the intra-layer.
3. Allowing each layer to read and write back LLRs in a different order.
4. Using the patch method based on variable-to-check messages.

To demonstrate the optimization process more intuitively, we compared the optimization structure of each step with the original decoding method. The original decoding method refers to the patch method based on LLR without any optimization in the processing order. We calculated the proportion of pipeline conflicts generated by the different decoding methods under different pipeline settings, and show it in Figure 11.

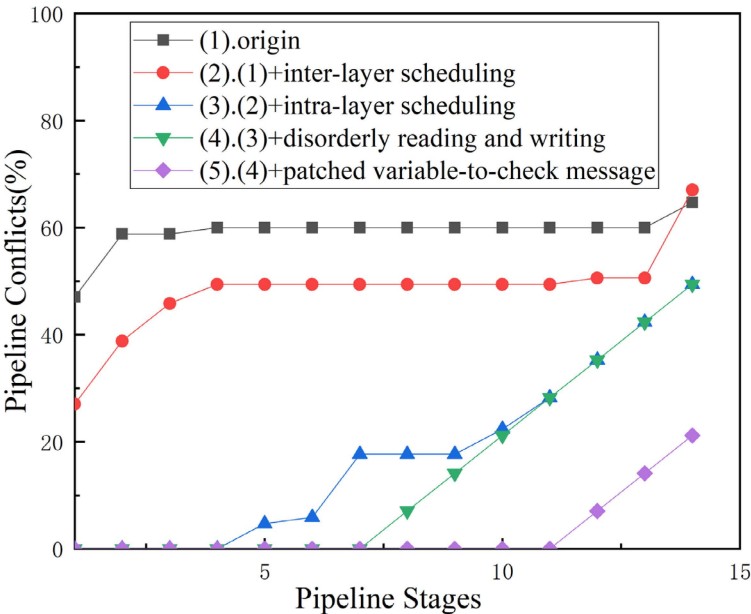

**Figure 11.** The pipeline conflicts comparison among different decoders.

In the original method, represented by line (1), when the number of pipeline stages exceeds 3, overlapping submatrices between adjacent layers in the PCM will all experience pipeline conflicts. Thus, the proportion of the pipeline remains stable at around 60%. As the number of pipeline stages gradually increases, it is also necessary to consider the possibility of conflicts between the two layers of the interval. When 14 pipeline stages are set, the proportion of pipeline conflicts further increases.

In line (2), a reordering of the inter-layer processing of decoding is used based on (1), resulting in a 10.59% reduction in the number of overlapping submatrices. It can be seen that roughly between the 4 and 11 pipeline stages, (2) reduces the pipeline conflict ratio by 10.59% compared to (1).

Line (3) reorders the intra-layer processing based on (2) to prevent overlapping submatrices from colliding with the pipeline when the number of pipeline stages is not set

too high. However, as the number of the pipeline stages approaches the row weight of the check matrix, the proportion of pipeline conflicts also approaches (1) and (2).

Line (4) allows each decoding layer to have a different order when reading LLR and writing LLR back to the RAM based on (3). This method solves the contradiction problem of read-and-write order rearrangement caused by overlapping submatrices with more than two consecutive layers connecting the same set of variable nodes.

Line (5) uses the patch method based on variable-to-check message applied on the basis of (4). According to formulae (10) (11) and the actual hardware structure in Figure 6, it can be inferred that the patch method based on variable-to-check message can reduce the number of pipeline conflicts for each layer the most, equal to the total number of pipeline stages for reading and writing LLR, as well as computing the variable-to-check message. In the practical design, LLR writing back to RAM and the reading out of RAM each require one pipeline stage. The calculation of the variable-to-check message requires one subtraction and one supersaturation process, assuming that a two-stage pipeline is required here. Therefore, curve (5) is equivalent to a horizontal shift of 4 units to the right compared to (4).

### 4.3. Comparison of Decoding Performance

To compare the decoding performance of the proposed architecture, we carried out an experiment on the priority-based with double update queue (PD) architecture in [22] and the residue-based scheme (RS) architecture in [19]. For a fair comparison, LLR, variable-to-check message, and check-to-variable message were quantified as 8-bit, 8-bit, and 6-bit in these three decoding architectures, respectively. And the decoders with three decoding architectures were all inserted into 11 pipeline stages. On the OFDM-PON platform, 1,000,000 OFDM frames were sent out for each received optical power (ROP) to calculate the BER. The BER curves of the decoder with proposed architecture, PD, and RS depending on the ROP are shown in Figure 12. The maximum number of decoding iterations was set to 10. It can be observed that with our proposed architecture, the decoder outperformed the decoder with PD by 0.125 dBm, and outperformed the decoder with RS by 0.375 dBm when the BER is $1 \times 10^{-6}$.

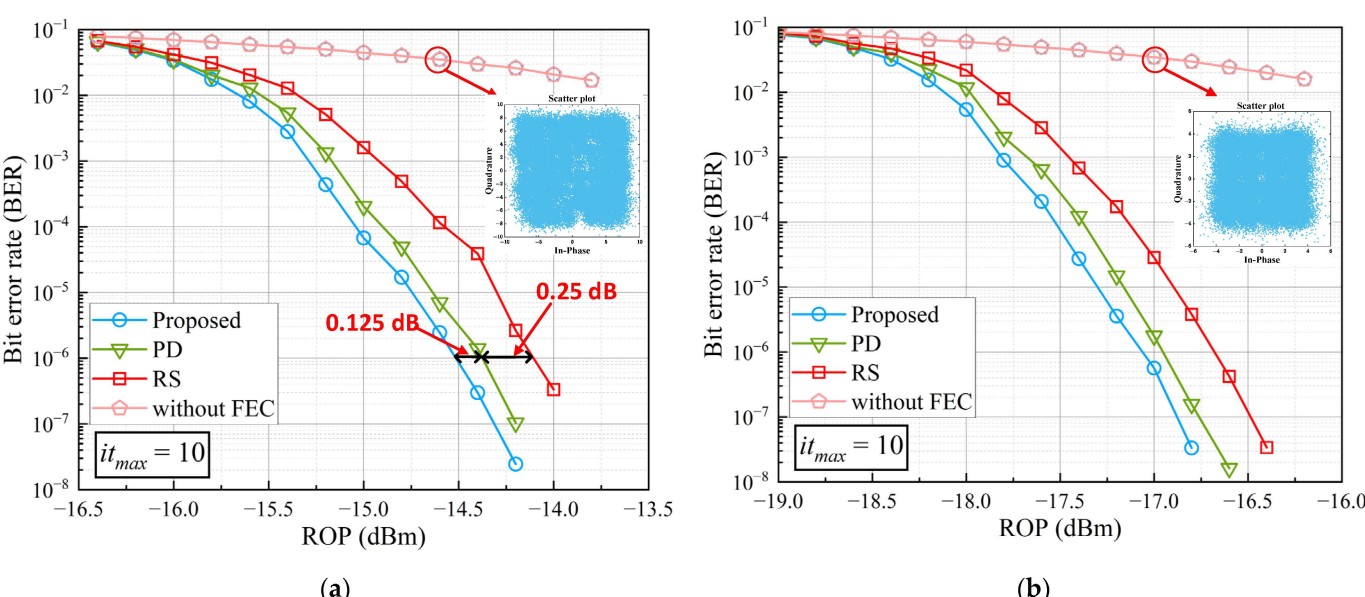

**Figure 12.** The BER comparison among decoders with different decoder architectures. (**a**) The modulation format is 64-QAM; (**b**) the modulation format is 16-QAM.

Further, we investigated the average iteration number among the three algorithms, as shown in Figure 13. The maximum iteration number was set to 10 and the iteration finished once the codeword $C$ and PCM $H$ satisfied $C \times H^T = 0$ or the decoding reached

the maximum iteration number. It can be seen that compared to our proposed architecture, the decoders based on PD and RS require an increase in the average number of iterations to achieve the same decoding performance.

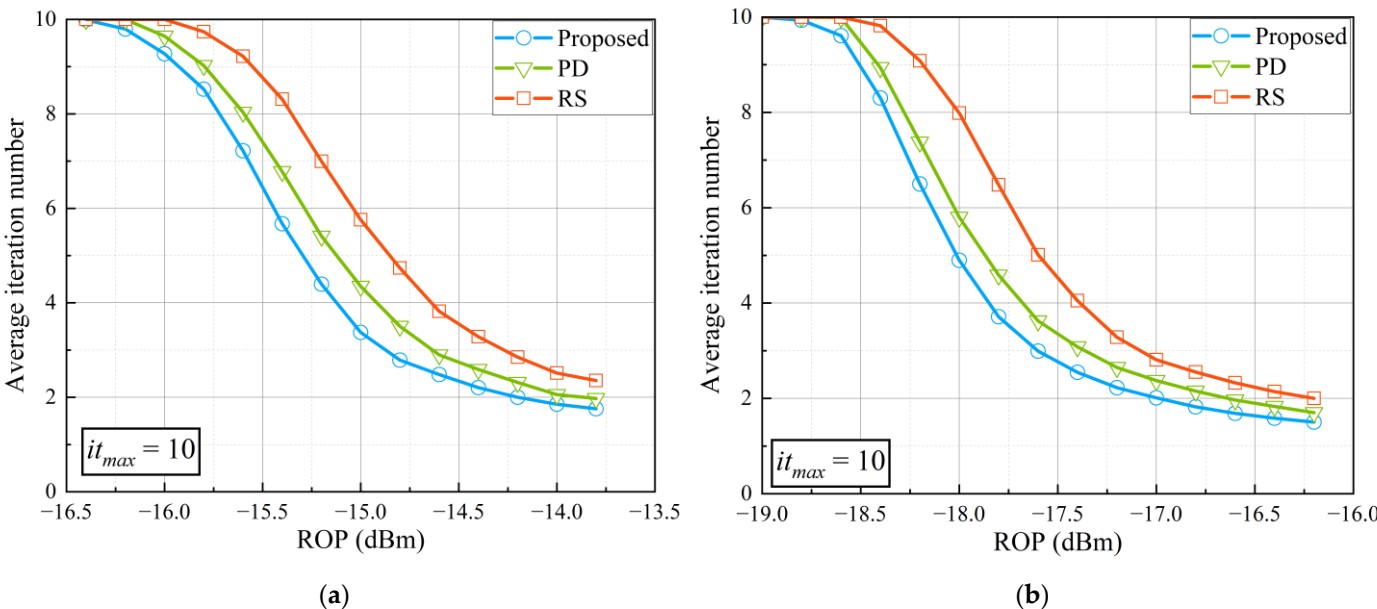

**Figure 13.** Average iteration number necessary for decoding with different architectures. (**a**) The modulation format is 64-QAM; (**b**) the modulation format is 16-QAM.

*4.4. Hardware Implementation*

As shown in Table 3, we conducted a comparison of the hardware resources utilized by the proposed decoding architecture with the priority-based architecture in our previous work [22] and the residue-based architecture in reference [19]. Compared with our previous work [22], the proposed architecture does not require two update queues, resulting in a slight reduction in LUTs and FFs. However, to ensure that each layer of the layered decoder can read and write LLRs in a different order during the decoding process, we replaced some FIFOs in other architectures with BRAM. Consequently, there was a slightly excessive use of BRAM.

**Table 3.** Implementation results for LDPC decoders with different architectures.

| Algorithm | Resource Utilization | | | $f_{max}$ [MHz] | $T_{norm}$ [Gbps] | HUE ($T_{norm}$/Resources) | | |
| | LUTs | FFs | 36 k BRAMs | | | Mbps/kLUT | Mbps/kFF | Mbps/BRAM |
|---|---|---|---|---|---|---|---|---|
| [19] | 40,700 | 26,925 | 40.5 | 142.8 | 10.8 | 265.3 | 401.1 | 266.7 |
| [22] | 26,744 | 19,594 | 27 | 310.0 | 8.2 | 306.3 | 418.5 | 303.7 |
| This work | 24,985 | 15,688 | 41 | 350.0 | 9.3 | 372.2 | 592.8 | 226.7 |

## 5. Conclusions

In this paper, we delve deeply into the challenges posed by pipeline conflicts in the hardware implementation of layered QC-LDPC decoders on the OFDM-PON platform. The main issue lies in the intricate data dependencies within the layered decoding process, which often lead to conflicts or stall cycles, thereby compromising the decoder's performance. To address this issue, we propose an optimized QC-LDPC decoder architecture, comprising four components: reordering of inter-layer processing, reordering of intra-layer submatrices, allowing for different read and write orders at each layer, and the patch method based on variable-to-check message. offline rearrangement of the processing order reduces the potential pipeline conflict problem to a limited extent. The patch method based on variable-to-check message further reduces conflicts because it allows for the moment of

utilization of gains during layer iterations to be delayed. And when pipeline problems are unavoidable, they can be compensated in time to minimize the performance loss.

For the rate 3/4 LDPC code of the IEEE802.16 standard, the traditional patch method based on LLR achieves a conflict rate of about 60% when inserting more than two pipeline stages. However, our proposed architecture can achieve the effect of no pipeline conflicts when inserting fewer than 11 pipeline stages. Compared with our previous work and the method in another paper, the sensitivity was improved by 0.125 dBm and 0.375 dBm, respectively, at a bit error rate of $10^{-6}$. And our proposed architecture also showed a significant reduction in the average number of iterations. We believe that the proposed LDPC decoding architecture can further improve the system performance of OFDM-PON. Additionally, the integration of the proposed decoder architecture with other advanced communication techniques, such as Multiple-Input Multiple-Output (MIMO) or adaptive modulation, could further boost the overall system performance. It is recommended that these aspects be explored in subsequent work.

**Author Contributions:** Conceptualization, Z.W. and J.Z.; methodology, Z.W. and Z.X.; software, Z.W.; validation, Z.W., K.C. and X.L.; formal analysis, Y.Q.; investigation, Y.L.; resources, J.Z.; data curation, K.C.; writing—original draft preparation, Z.W.; writing—review and editing, Z.W., J.Z., Y.Q. and Y.L.; supervision, J.Z. All authors have read and agreed to the published version of the manuscript.

**Funding:** This work was supported in part by the National Key Research and Development Program of China (2021YFB2900800); Science and Technology Commission of Shanghai Municipality (22511100902, 22511100502); 111 Project (D20031).

**Institutional Review Board Statement:** Not applicable.

**Informed Consent Statement:** Not applicable.

**Data Availability Statement:** Data are contained within the article.

**Conflicts of Interest:** Author Xiaoqun Liu was employed by the company Beijing Smartchip Microelectronics Technology Co., Ltd. The remaining authors declare that the research was conducted in the absence of any commercial or financial relationships that could be construed as a potential conflict of interest.

**Acronyms**

| | |
|---|---|
| LDPC | Low-Density Parity-Check |
| QC-LDPC | Quasi-Cyclic Low-Density Parity-Check |
| TDM-PON | Time Division Multiplexing Passive Optical Network |
| WDM-PON | Wavelength Division Multiplexing Passive Optical Network |
| OFDM-PON | Orthogonal Frequency Division Multiplexing Passive Optical Access Network |
| PAPR | Peak-to-Average Power Ratio |
| FEC | Forward Error Correction |
| PCM | Parity-Check Matrix |
| LLR | Log-Likelihood Ratio |
| MSA | Min-Sum Algorithm |
| OMSA | Offset Min-Sum Algorithm |
| NMSA | Normalized Min-Sum Algorithm |
| FIFO | First In First Out |
| RAM | Random Access Memory |
| DFB-LD | Distributed Feedback Laser |
| SSMF | Standard Single-Mode Fiber |
| OA | Optical Attenuator |
| BER | Bit Error Rate |
| ROP | Received Optical Power |
| MIMO | Multiple-Input Multiple-Output |

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
