# Peer review of "Efficient Pipeline Conflict Resolution for Layered QC-LDPC Decoders in OFDM-PON"

_photonics, doi:10.3390/photonics11050429_

Round 1

Reviewer 1 Report

Comments and Suggestions for Authors

This paper addresses the issue of pipeline conflicts within the hardware implementation of layered Quasi-Cyclic Low-Density Parity-Check (QC-LDPC) decoders, a critical component in Orthogonal Frequency Division Multiplexing-Passive Optical Network (OFDM-PON) systems. The results reveal that the proposed architecture yields notable improvements in sensitivity --specifically, enhancements of 0.125 dBm and 0.375 dBm when compared to previous works and another referenced method, respectively. Overall, the paper is interesting to the readers. Authors are advised to address the following comments in the revised version:

1-    It is recommended to enhance the abstract to better articulate the study's innovation, methodology, research findings, and significance. A clear and concise abstract will help to immediately convey the value and novelty of the research to readers.

2-    Write you key objectives in 3 points, and paper organization at the end of Section 1.

3-    Provides all keywords in the beginning of the Section 1 such as QC-LDPC, OFDM-PON, FEC, etc., before using them in the text.

4-    A detailed comparison table of the existing schemes must be added in Section 1.1 Also discuss about various optical network studies with their strength and weakness in Table 1:  -Butt, Rizwan Aslam, et al. "A survey of dynamic bandwidth assignment schemes for TDM-based passive optical network." Journal of Optical Communications (2020).

-Mohammadani, Khalid H., et al. "A QoS provisioning architecture of fiber wireless network based on XGPON and IEEE 802.11 ac." Journal of Optical Communications (2021).

-Ashraf, M. W., et al. "Disaster-resilient optical network survivability: a comprehensive survey." PhotonicsVol. 5. No. 4. MDPI, 2018.

5.     Rewrite Eq. 6 and 11 by following the proposed scheme.

6.     Enhancement of the readability of Figures 2-4, as the current text size makes them challenging to interpret. Clear and legible figures are crucial for conveying complex information effectively.

7.     Please add a pseudo code of the proposed scheme in a standard format, with a detailed definition of all mathematical terms and symbols used, to facilitate comprehension and replication of the proposed model (on page 4).

8.     I recommend to verify the parameter values used in Tables 1.

9.     Section 5 should be enhanced by discussing the existing issues, solutions, and future research directions in detail.

Comments on the Quality of English Language

Minor

Author Response

Thank you for your comments. Please see the attachment for all responses.

Reviewer 2 Report

Comments and Suggestions for Authors

The authors solved the problem of pipeline conflicts arising from data dependencies in the hardware implementation of layered QC-LDPC decoders. They proposed an efficient layered decoding architecture. The performance of the architecture was validated by experiment. Sensitivity improvement were observed in the experiments compared with previous work and other work.  This manuscript contains novel and interesting solution and experimental results. These results have been compared with the previous results. The paper is clearly written and well organized. The steps of the proposed solution were described clearly. The paper can be interesting for the researchers and engineers occupied in optical communications.

The manuscript can be accepted for publication in the present form without any revision.

Author Response

(The authors gave the same response as above.)

Reviewer 3 Report

Comments and Suggestions for Authors

Evaluation.

      The study proposes  a QC_LDPC decoding structure based on the patch method (on n  variable-to-check message) to improve the OFDM-PON system and solve the pipeline  conflicts problem; The document was well presented and discussed. However, several points deserve to be better discussed.

1/*-The authors have not provided details and description of the proposed system (the organization chart, complete transmission and reception chain).

2/*- In the OFDM signal part, it is necessary to provide the mathematical details used for the overall system.

3/*- explain and detail the role of the PON in the proposed system ( mathematical calculations are missing)

4/*- Also for the part of the LDPC technique, give the mathematical details.

5/*- you use the hybrid optical system (PON - OFDM), give the analysis and evaluation of this system.

6/*- As long as you study the optical system, I ask you to use the optysistem software to validate your results and draw the eye diagram plus the EVM.

7/*- Add more recent literature work related to this work.

     Finally, I propose a major revision

Comments on the Quality of English Language

Moderate editing of English language required

Author Response

(The authors gave the same response as above.)

Round 2

Reviewer 1 Report

Comments and Suggestions for Authors

The authors have addressed our comments in the revised version. Therefore, we are happy to accept this paper in its current form. 

Comments on the Quality of English Language

Minor